# Core-Shell-Shell Upconversion Nanomaterials Applying for Simultaneous Immunofluorescent Detection of Fenpropathrin and Procymidone

**DOI:** 10.3390/foods12183445

**Published:** 2023-09-15

**Authors:** Yang Song, Jingyi Jin, Liuling Hu, Bingqian Hu, Mengyao Wang, Lilong Guo, Xiyan Lv

**Affiliations:** Tianjin Key Laboratory of Animal and Plant Resistance, Tianjin Normal University, Tianjin 300387, China; 18920492831@163.com (J.J.); 17622730132@163.com (L.H.); 18993569506@163.com (B.H.); f18300702684@163.com (M.W.); 13502124216@163.com (L.G.); 18102185913@163.com (X.L.)

**Keywords:** core-shell-shell, magnetic separation, fenpropathrin, procymidone

## Abstract

This study synthesized the NaGdF_4_@NaGdF_4_: Yb, Tm@NaGdF_4_: Yb, Nd upconversion nanoparticles (UCNPs), combined with another three-layer structure NaYF_4_@NaYF_4_: Yb, Er@NaYF_4_ UCNPs, with a core-shell-shell structure, effectively suppressing fluorescence quenching and significantly improving upconversion luminescence efficiency. Two types of modified UCNPs were coupled with antibodies against fenpropathrin and procymidone to form signal probes, and magnetic nanoparticles were coupled with antigens of fenpropathrin and procymidone to form capture probes. A rapid and sensitive fluorescence immunoassay for the simultaneous detection of fenpropathrin and procymidone was established based on the principle of specific binding of antigen and antibody and magnetic separation technology. Under the optimal competitive reaction conditions, different concentrations of fenpropathrin and procymidone standards were added to collect the capture probe-signal probe complex. The fluorescence values at 542 nm and 802 nm were measured using 980 nm excitation luminescence. The results showed that the detection limits of fenpropathrin and procymidone were 0.114 µg/kg and 0.082 µg/kg, respectively, with sensitivities of 8.15 µg/kg and 7.98 µg/kg, and they were applied to the detection of fenpropathrin and procymidone in tomatoes, cucumbers, and cabbage. The average recovery rates were 86.5~100.2% and 85.61~102.43%, respectively, with coefficients of variation less than 10%. The results showed good consistency with the detection results of high-performance liquid chromatography, proving that this method has good accuracy and is suitable for the rapid detection of fenpropathrin and procymidone in food.

## 1. Introduction

Upconversion nanomaterials (UCNPs) have excellent optical properties, which can convert long-wavelength near-infrared light into short-wavelength visible light [1]. Compared with traditional fluorescent dyes such as organic dyes, quantum dots, and semiconductor materials, they have good chemical stability, strong penetration, low background fluorescence value, and low toxicity [2,3]. However, the specific surface area of upconversion nanomaterials is large and prone to fluorescence quenching, resulting in a significant decrease in luminescence efficiency [4,5]. By continuing to coat the outer shell on the surface of the core to construct the core-shell structure of UCNPs, it can effectively reduce its surface defects and isolate the external unfavorable factors to improve the upconversion luminescence efficiency [6]. Currently, upconversion nanomaterials have been widely used in fields such as biological detection, biological imaging, biosensing, and immune analysis [7,8]. With the increasing severity of food safety issues, in order to improve the speed, sensitivity, and accuracy of detection, UCNPs have been applied to the detection of pesticide residues, mycotoxins, heavy metals, and other pollutants due to their unique advantages [9,10]. Qin et al. [11] prepared two types of core-shell UCNPs, NaYF_4_: Yb, Er@NaYF_4_ and NaYF_4_: Yb, Tm@NaYF_4_, and coupled them with two aptamers, OTA and ZEN, respectively, for the preparation of upconversion fluorescent probes, and established a magnetically controlled two-colour upconversion fluorescence method for the simultaneous determination of ochratoxin A (OTA) and zearalenone (ZEN) in maize and oat flour. Compared with other methods, this method is simple, sensitive, and specific. Su et al. [12] synthesized NaErF_4_: 0.5% Tm^3+^@NaYF_4_ core inert shell UCNPs and prepared a UCNPs/polydopamine system, which can achieve quantitative analysis and detection of carbaryl in 11 tea samples, making it more accurate and reliable compared to traditional methods.

Fenpropathrin is a broad-spectrum pyrethroid insecticide and acaricide [13], which has the characteristics of high efficiency, low toxicity, low residue, and can be degraded, etc. In order to control pests and improve the yield of crops, fenpropathrin is often widely used in agricultural pest control [14]. Procymidone is a new low-toxicity fungicide of the organochlorine class, which is mainly used for the control of gray mold and botrytis in fruits and vegetables [15,16]. Studies have shown that fenpropathrin can accumulate in various organs of the human body through enrichment, seriously interfere with the nerve and immune system, and cause severe effects on human health, such as dizziness, headache, nausea, etc. [17]. Long-term excessive consumption of procymidone will accumulate quantitatively in the human body, seriously endangering the human nervous system and reproductive organs [18]. These two pesticide residues have become routine detection items in fruits and vegetables [19]. As two common pesticides found in pyrethroids and organochlorines, fenpropathrin and procymidone are often detectable simultaneously, and pesticide residue analysis often uses instrument methods [20,21], such as high-performance liquid chromatography, gas chromatography, etc. These instrument methods have high sensitivity, strong specificity, and a wide range of applications. However, the instrument equipment is expensive, there is a lot of preparation work in the early stage of the experiment, and the operation process is relatively complex, requiring professional personnel to operate, which is difficult to accommodate during on-site rapid detection. It is necessary to develop a new technology that is more rapid, efficient, and sensitive for the simultaneous detection of fenpropathrin and procymidone.

In this experiment, a three-layer upconversion nanomaterial was synthesized using the thermal decomposition method, while also being combined with an existing three-layer upconversion nanomaterial in our laboratory. Two types of UCNPs were surface-modified to increase their dispersibility. The two modified UCNPs were coupled with fenpropathrin and procymidone antibodies to prepare signal probes, and magnetic nanoparticles were coupled with fenpropathrin and procymidone antigens to prepare capture probes. Under optimal reaction conditions, the capture probe competes with the pesticide standard to bind the signal probe. The capture probe-signal probe complex is separated and collected through an external magnetic field, and its fluorescence intensity is measured using 980 nm external excitation luminescence. A fluorescence immunoassay method for the simultaneous detection of fenpropathrin and procymidone is established. The method is rapid and convenient, sensitive and efficient, which is suitable for the detection of fenpropathrin and procymidone in foodstuffs simultaneously.

## 2. Materials and Methods

### 2.1. Materials and Instruments

Fenpropathrin, procymidone, GdCl_3_·6H_2_O, YbCl_3_·6H_2_O, TmCl_3_·6H_2_O, NdCl_3_·6H_2_O, tetraethoxysilane (TEOS, 98%), 3-aminopropyltriethoxysilane (APTES, 98%), and ammonium fluoride were purchased from Adamas (Shanghai, China). Oleic acid, 1-octadecene, methanol, ethanol, cyclohexane, sodium borohydride, Tween-20, anhydrous sodium sulfate, N-propylethylenediamine (PSA), and activated carbon were purchased from Kmart (Tianjin) Chemical Technology Co., Ltd. (Tianjin, China). Fenpropathrin monoclonal antibody, fenpropathrin antigen, procymidone monoclonal antibody, and procymidone antigen were purchased from Shandong Landu Biotechnology Co., Ltd. (Jinan, China). Sodium hydroxide, acetonitrile, and sodium chloride were purchased from Tianjin Chemical Reagent Supply and Marketing Company, Tianjin, China. Ammonia and BCA kits (Bicinchoninic Acid) were purchased from Tianjin Yuanli Chemical Co., Ltd. (Tianjin, China). Glutaraldehyde was purchased from Shanghai Meryer Biochemical Technology Co., Ltd. (Shanghai, China). Bovine serum albumin (BSA, 98%) was purchased from Merck, Darmstadt, Germany. Aminated polystyrene magnetic microspheres (MNPs, 0.5% *w*/*v*) were purchased from Beijing Baseline Co., Ltd. (Tianjin, China). Argon was purchased from Liquefied Air (Tianjin) Co., Ltd. (Tianjin, China). NaYF_4_@NaYF_4_: Yb, Er@NaYF_4_ UCNPs were synthesized by thermal decomposition method in our laboratory [22]. The morphology and size of UCNPs were determined by FEI TECNAI G20 transmission electron microscopy (TEM, FEI Company, Hillsboro, OR, USA). Fourier transform infrared spectroscopy (FTIR) of UCNPs was determined by an FTIR spectrophotometer (Perkin Elmer, Waltham, MA, USA). The powder X-ray diffraction (XRD) measurement of UCNPs was performed by the AXIS-ULTRA-DLD instrument (Millipore, New York, NY, USA). The fluorescence intensity of the upconversion nanoparticles was detected by an F-2500 fluorescence spectrophotometer equipped with a 980 nm laser excitation device. Waters e2695 HPLC was from Thermo Fisher Scientific, Waltham, MA, USA. All chemicals used are analytical grade.

### 2.2. Synthesis of NaGdF_4_@NaGdF_4_: Yb, Tm@NaGdF_4_: Yb, Nd UCNPs

NaGdF_4_@NaGdF_4_: Yb, Tm@NaGdF_4_: Yb, Nd UCNPs were prepared by the high-temperature thermal decomposition method. Firstly, mononuclear NaGdF_4_ UCNPs were prepared. A total of 2 mmol GdCl_3_·6H_2_O, 30 mL 1-octadecene, and 12 mL oleic acid were added to the three-port flask in turn and heated to 160 °C in high purity argon and maintained for 60 min. When the reactant was cooled to room temperature, the 20 mL methanol solution dissolved with 5 mmol NaOH and 8 mmol NH_4_F was added drop by drop into three bottles, stirred at room temperature for 30 min, and then heated to 100 °C for 20 min, so that the methanol is fully evaporated. The temperature was raised to 300–310 °C again and kept for 1–2 h. The reaction system was cooled to room temperature, centrifuged at 8500 rpm (8000× *g*) for 10 min, washed several times with ethanol and cyclohexane, alternately, dried in an oven at 60 °C, and collected for use. NaGdF_4_ UCNPs were obtained.

The core-shell NaGdF_4_@NaGdF_4_: Yb, Tm UCNPs were further synthesized. A total of 2 mmol of ReCl_3_·6H_2_O (Re: 10% mol Gd^3+^, 89% mol Yb^3+^, 1% mol Tm^3+^) was put into a three-port flask, and 30 mL of octadecene and 12 mL of oleic acid were added. The reaction was heated to 160 °C for 60 min in high-purity argon and cooled to room temperature. Next, we added 2 mmol NaGdF_4_ nucleus dissolved in 3 mL cyclohexane dropwise and stirred for 20 min. Then 20 mL methanol solution containing 5 mmol NaOH and 8 mmol NH_4_F was added and heated to 100 °C and 310 °C, respectively. NaGdF_4_@NaGdF_4_: Yb, Tm UCNPs were obtained after cleaning and drying.

Finally, NaGdF_4_@NaGdF_4_: Yb, Tm@NaGdF_4_: Yb, Nd UCNPs was synthesized. A total of 2 mmol of ReCl_3_·6H_2_O (Re: 80% mol Gd^3+^, 10% mol Yb^3+^, 10% mol Nd^3+^), 30 mL of 1-octadecene, and 12 mL of oleic acid were added to the three-port flask in turn, and the temperature was raised to 160 °C for 60 min. Cooling to room temperature, 2 mmol NaGdF_4_@NaGdF_4_: Yb, Tm dissolved in 3 mL cyclohexane was added dropwise and stirred for 20 min. A 20 mL methanol solution dissolved with 5 mmol NaOH and 8 mmol NH_4_F was added, and the subsequent steps were the same as above. Finally, NaGdF_4_@NaGdF_4_: Yb, Tm@NaGdF_4_: Yb, Nd UCNPs with a three-layer structure of inert core/active shell/active shell was obtained.

### 2.3. Modification of Upconversion Nanomaterials

The surface of upconversion nanomaterials prepared using the above method contained oleic acid molecules, which showed hydrophobicity, and to have good dispersion in an aqueous solution, the surface of UCNPs was modified. Weighed 0.1 g UCNPs ultrasonically dispersed in 50 mL ethanol after rapid stirring for 30 min, followed by vigorous stirring at 45 °C, adding 4 mL ammonia and 30 mL secondary water, and adding 40 µL tetraethyl silicate to the mixture within 5 h, followed by the dropwise addition of 100 µL APTES solution to the reaction for 5 h. After the reaction was completed, the white solid precipitate was dried to obtain UCNPs@SiO_2_@NH_2_.

### 2.4. Preparation and Optimization of Capture Probe and Signal Probe

Capture probes (signal probes) were prepared using the classical glutaraldehyde cross-linking method. A total of 10 mg MNPs (UCNPs@SiO_2_@NH_2_) were dissolved in 5 mL 10 mmol/L PBS solution, sonicated at 40 kHz for 30 min, and then added 1.25 mL 25% glutaraldehyde solution and 0.1 g sodium borohydride, slowly shaking at 500 rpm for 1 h at room temperature. After the reaction, the precipitate was collected by magnetic field separation (centrifugation) centrifuged at 8500 rpm (8000× *g*) for 5 min. The precipitate was washed three times with PBS and dispersed in 5 mL PBS. A certain amount of antigen (antibody) was added to the solution after sonication at a frequency of 40 kHz for 10 min. After shaking at 500 rpm for 6 h under room temperature, the precipitate was washed three times. Then the precipitate was dispersed in 5 mL PBS (containing 1% BSA). After 6 h of reaction, the precipitate was collected and dispersed in 5 mL PBS, and stored at 4 °C.

The amount of antigen (antibody) added to the capture probe (signal probe) during the preparation process was optimized. We added 20 µg, 40 µg, 60 µg, 80 µg, 100 µg, 120 µg of antigen (antibody), respectively, collected the supernatant at the end of the reaction, and used the BCA kit to measure the content of antigen (antibody) in the supernatant liquid after the reaction, which can determine the content of the coupled protein in the reaction. We then calculated the coupling rate according to the following formula, so as to determine the optimal amount of antigen (antibody) added.
$$\mathrm{coupling}\;\mathrm{rate}\;(\%)=\frac{Mc}{Mt}\times 100\%$$

Among them: *Mc* is the amount of antibody (antigen) coupling and *Mt* is the total amount of antibody (antigen) added.

### 2.5. Optimization of the Addition Amount of Capture Probes

The addition amount of the two signal probes was fixed at 200 µL, and the addition amount of the two capture probes was set to 40 µL, 60 µL, 80 µL, 100 µL, 120 µL, and 140 µL, respectively, and the reaction was carried out at room temperature for 1 h. After the reaction, the signal probe-capture probe complex was collected by an external magnetic field, washed three times with PBS, and the complex was dispersed in 2 mL PBS to determine its fluorescence value.

### 2.6. Establishment of Upconversion Magnetic Separation Immunofluorescence Detection Method

A total of 100 µL of each of the two capture probes was mixed with 50 µL of each of the different concentrations of fenpropathrin and procymidone in a 5 mL centrifuge tube, and 200 µL of each of the two signal probes was added, and reaction was carried out by shaking at 500 rpm for 1 h under room temperature. After the reaction, the signal probe-capture probe complex was collected by magnetic separation, and the precipitate was washed three times with PBST solution. Then the complex was dispersed in 2 mL PBST solution. The fluorescence intensity values at 542 nm and 802 nm were measured by a fluorescence spectrophotometer with a 980 nm semiconductor laser emitter, and the immunofluorescence standard curves at different target concentrations were established. In the competitive reaction system, the buffer system (PBS solution, PBST solution, PBST-milk powder solution) was determined, the tween-20 content (0.01%, 0.05%, 0.1%, 0.5%), the methanol content (0%, 2.5%, 5%, 10%), and the pH value (5, 6, 7, 8, and 9) in the buffer solutions were optimized, and the competitive reaction times were set (40 min, 50 min, 60 min, 70 min, 80 min). These were established in order to determine the optimal buffer system, tween-20 content, methanol content, and pH value by comparing the IC_50_ and F_max_/IC_50_ values, respectively. (F_max_ is the maximum value of fluorescence and IC_50_ is the corresponding standard concentration when the inhibition rate is 50%, that is, sensitivity).

### 2.7. Method Specificity

To evaluate the specificity of the method, some structural analogues of fenpropathrin and procymidone and other common pesticides, namely fenvalerate, cypermethrin, dicofol, carbendazim, imidacloprid, and acetochlor, were selected for cross-reaction experiments. The concentration of each standard was 10 µg/L. According to the immunoassay established in this experiment, the capture probe and the signal probe were added in turn. The fluorescence values at the emission wavelengths of 542 nm and 802 nm after the reaction were recorded as I and the fluorescence values after the reaction of only the capture probe and the signal probe were recorded as I_0_ to calculate ΔI (ΔI = I_0_ − I).

### 2.8. Actual Sample Testing

The pretreatment method of samples for high-performance liquid chromatography detection refers to GB/T 20769-2008 [23]. Three samples of tomato, cucumber, and cabbage were extracted and purified for HPLC detection. The samples used for upconversion immunoassay were homogenized and extracted with acetonitrile. A total of 100 mg PSA was added to the extract, 50 mg activated carbon was added to tomatoes and cucumbers, and 75 mg activated carbon was added to cabbage. After shaking and mixing, the supernatant was centrifuged at 8500 rpm (8000× *g*) for 5 min and filtered through a 0.22 µm filter membrane, evaporated to near dryness at 50 °C, and redissolved with 2 mL methanol to be measured.

Chromatographic conditions: chromatographic column: Bridge C18 (5 µm, 4.6 × 250 mm); mobile phase: methanol–water (32:68); flow rate: 1.0 mL/min; column temperature: 25 °C; injection volume: 20 µL; detection wavelength: 204 nm.

## 3. Results and Discussion

### 3.1. Characterization of Upconversion Nanomaterials

The structure of nanoparticles determines their performance. The main factors affecting the formation of nanoparticle structure are its composition, synthesis temperature, and reaction time. NaGdF_4_ UCNPs were synthesized by a high-temperature thermal decomposition method. After the evaporation of methanol, the temperature was increased to 300 °C and 310 °C, and the reaction time was set to 1 h, 1.5 h, and 2 h, respectively. The morphological characteristics of NaGdF_4_ UCNPs synthesized under different conditions were characterized by transmission electron microscopy. The results are shown in Figure 1a–f. The morphologies of nanocrystals synthesized at different temperatures and times were significantly different. Figure 1a–c is the electron microscope image of NaGdF_4_ UCNPs obtained at 300 °C for 1 h, 1.5 h, and 2 h, respectively. Under the condition of 300 °C, the reaction time is extended from 1 h to 2 h, and the obtained NaGdF_4_ nanocrystals have a tendency to become larger, but there are uneven shape and size, crystal agglomeration, and poor dispersion. Raising the temperature to 310 °C, Figure 1d–f is the electron microscope image of NaGdF_4_ UCNPs obtained at 310 °C for 1 h, 1.5 h, and 2 h, respectively. As shown in Figure 1e, when the reaction conditions are 1.5 h and 310 °C, the obtained nanocrystals have good dispersibility, but the shape and size are uneven. As we continued to extend the reaction time to 2 h, as shown in Figure 1f, the obtained nanocrystals had regular morphology, uniform size, and good dispersibility. The particle size is about 20 nm, so the optimal reaction conditions are set to 2 h and 310 °C. From the results of synthesis optimization, it can be seen that the change in heating temperature has the greatest influence on the synthesized nanomaterials, and the precise time and temperature are very important for the synthesized materials.

Figure 2a,b shows NaGdF_4_@NaGdF_4_: Yb, Tm UCNPs and NaGdF_4_@NaGdF_4_: Yb, Tm@ NaGdF_4_: Yb, Nd UCNPs synthesized under the optimal reaction conditions. The particle sizes are 39 nm and 50 nm, respectively. The diameter of nanocrystals increases layer by layer, indicating that the core-shell coating is successful. To increase the dispersion of upconversion nanomaterials, the surface of the synthesized core-shell-shell upconversion nanoparticles was modified. From Figure 2c, it can be seen that a thin silicon shell appears on the surface of the modified material, and the particle size is about 58 nm.

Figure 3a is the fluorescence spectrum of NaGdF_4_@NaGdF_4_: Yb, Tm@NaGdF_4_: Yb, Nd UCNPs. Under the excitation of the 980 nm wavelength, the nanomaterial has a maximum characteristic emission peak at 802 nm. As shown in Figure 3b, NaYF_4_@NaYF_4_: Yb, Er@NaYF_4_ UCNPs and NaGdF_4_@NaGdF_4_: Yb, Tm@NaGdF_4_: Yb, Nd UCNPs were mixed for fluorescence determination. It was found that the maximum characteristic emission peaks of the two materials existed independently, the maximum characteristic emission peaks appeared at 542 nm and 802 nm, respectively, and the peaks were in the same order of magnitude. Therefore, they can be used to label two targets simultaneously for detection studies.

It can be seen from the fluorescence spectrum characterization that both UCNPs with three-layer structures have superior luminescence efficiency, and the fluorescence intensity is increased by several times compared with the corresponding two-layer structure UCNPs. The outermost layer of NaYF_4_@NaYF_4_: Yb, Er @NaYF_4_ UCNPs is coated with an inert shell, which can effectively reduce the surface defects of nanomaterials and reduce the interference of the external environment, thereby preventing fluorescence quenching. The active ions of NaGdF_4_ @ NaGdF_4_: Yb, Tm@NaGdF_4_: Yb, Nd UCNPs are effectively confined to the luminescent layer, reducing the energy loss caused by its aimless migration. At the same time, the doping of sensitizer Yb and Nd ions in the shell improves the absorption of excitation light by nanocrystals, so the luminescence efficiency of upconversion nanomaterials is significantly enhanced. Both UCNPs can be applied to fluorescence immunoassay, which greatly improves the efficiency of detection.

The crystal structure was identified by X-ray diffraction. The NaGdF_4_@NaGdF_4_: Yb, Tm@ NaGdF_4_: Yb, Nd identification results are shown in Figure 3c. The diffraction peak is consistent with the hexagonal β-phase UCNP (JCPDS No.27-0699) standard map comparison results, indicating that the upconversion nanomaterial is a hexagonal phase crystal and has good luminous efficiency.

The modified upconversion nanomaterials were characterized by Fourier transform infrared spectroscopy. As shown in Figure 3d, the silicon–oxygen symmetric stretching vibration of NaGdF_4_@NaGdF_4_: Yb, Tm@NaGdF_4_: Yb, Nd@SiO_2_@NH_2_ appears in the region of about 1066 cm^−1^, indicating that the material has been coated with a silicon shell; the stretching and bending vibration of the amino group appeared at 3134 cm^−1^, indicating that the surface of the material had an amino group and could be used to couple biological macromolecules.

### 3.2. Optimization of Antigen and Antibody Addition Amount

Since the amount of antigen and antibody added will affect the binding amount of the capture probe and the signal probe and the sensitivity of the method, during the preparation of the probe, the BCA kit was used to determine the concentration of the uncoupled antigen (antibody) in the supernatant, and the coupling amount and coupling rate were calculated to optimize the amount of antigen and antibody added to the probe. As shown in Figure 4b, the addition amount of fixed modified UCNPs is 10 mg. When the addition amount of fenpropathrin antibody is 40–100 µg, the coupling amount and coupling rate increase with the increase of antibody addition amount. When the addition amount is 100 µg, the coupling rate of antibody and UCNPs reaches the maximum value. As we continued to increase the addition amount of antibody, the coupling amount did not increase, and the coupling rate decreased, indicating that the binding site of UCNPs had reached saturation. From the perspective of saving antibodies, 100 µg of fenpropathrin antibody was chosen as the optimal addition amount for the preparation of signal probe, at which time the coupling rate was 77.56%. As shown in Figure 4c, the addition amount of fixed MNPs was 2 mL, and the highest coupling rate of 73.9% was achieved when the addition amount of fenpropathrin antigen was 80 µg. Similarly, according to Figure 4d,e, the optimal addition amount of procymidone antibody was finally determined to be 10 µg with a coupling rate of 69.32%, and the optimal addition amount of procymidone antigen was 80 µg with a coupling rate of 81.96%.

### 3.3. Optimization of the Amount of Capture Probe Addition

The amount of capture probe directly affects the binding effect of the antigen–antibody, thus affecting the fluorescence value of the reaction system. As shown in Figure 4f, the addition amount of the two signal probes was fixed at 200 µL. When the addition amount of the two capture probes is 40–100 µL, the fluorescence value of the capture probe-signal probe complex increases gradually after the reaction. When the addition amount of the two capture probes is 100 µL, the fluorescence value of the reaction system reaches the maximum. Continuing to increase the amount of capture probes will interfere with the reaction system and reduce the fluorescence value. Therefore, the optimal addition amount of the two capture probes is 100 µL.

### 3.4. Establishment of Upconversion Magnetic Separation Immunofluorescence Detection Method

It is possible to determine the optimal competitive reaction system by comparing the IC_50_ value and F_max_/IC_50_ value. First, the best buffer system is selected, as shown in Figure 5a,b. When PBST solution is used as the buffer system, the IC_50_ value is the lowest and the F_max_/IC_50_ value is the highest. Compared with the other two buffer systems, the sensitivity is higher, so PBST is selected as the buffer system to continue optimization. PBST solution is mainly a mixed system of PBS and Tween-20, and a certain amount of Tween-20 can help to reduce nonspecific adsorption in the reaction, but too much of it can affect antigen-antibody binding [24]. According to Figure 5c,d, with the increase of Tween-20 content, IC_50_ decreased first and then increased. When the Tween-20 content was 0.05%, the IC_50_ value was the lowest, and the F_max_/IC_50_ value was the highest. Therefore, the Tween-20 content in the PBST solution was 0.05%. As shown in Figure 5e,f, IC_50_ decreased first and then increased with the increase of pH value. When the pH value was 7–7.4, the IC_50_ values of the two pesticides were lower, and the F_max_/IC_50_ value was higher. Therefore, the pH value of the control system in the competitive reaction was between 7 and 7.4. Methanol content in the buffer system affects the binding of chemical bonds between antigens and antibodies. When the methanol content is 2.5%, the IC_50_ value is the lowest and the F_max_/IC_50_ value is the highest. Therefore, the buffer system was determined to be a PBST solution with a Tween content of 0.05%, a methanol content of 2.5%, and a pH value of 7–7.4. The reaction time will affect the binding of antigens and antibodies, thereby affecting the fluorescence value of the reaction results. When the competitive reaction lasts for 1 h, the fluorescence value of the reaction system is the highest and the reaction effect is the best.

Based on the above optimal competitive reaction conditions, the standard curves for the detection of fenpropathrin and procymidone were established by taking the logarithmic values of the concentrations of fenpropathrin and procymidone as the horizontal coordinates, and the fluorescence change values ΔI (ΔI = I_0_ − I, and I and I_0_ are the fluorescence values of the system with and without the standard, respectively) of the reaction system at 542 nm and 802 nm under the different concentrations of the standard products as the vertical coordinate, respectively. The upconversion fluorescence changes of different concentrations of fenpropathrin and procymidone standards are shown in Figure 5j. With the increase of the concentration of the two pesticide standards, the fluorescence value of the reaction system gradually decreases. According to Figure 5k,l, the limits of detection (LODs, IC_15_) were 0.114 μg/kg and 0.082 μg/kg for fenpropathrin and procymidone, and the sensitivities were 8.15 μg/kg and 7.98 μg/kg, respectively. (IC_15_ is the standard concentration corresponding to the inhibition rate of 15%, that is, the limit of detection). The linear range of fenpropathrin and procymidone was 0.02~62.5 µg/kg.

### 3.5. Method Specificity

Figure 6 shows the fluorescence changes (ΔI) at 542 nm and 802 nm of the reaction system when different pesticides are added. Evaluate the specificity of this method by comparing fluorescence changes. When the standards of fenpropathrin and procymidone were added, the fluorescence value of the reaction system changed significantly, indicating that the two signal probes coupled with antibodies could specifically bind to fenpropathrin and procymidone. When other types of pesticide standards are added, the fluorescence value of the reaction system changes slightly, with only some non-specific adsorption, indicating that the method has high specificity.

### 3.6. Matrix Impact Elimination and Addition Recovery Experiment

Vegetables have more complex components, such as pigments, proteins, and vitamins, which may interfere with the test results. Dilution of the extract after sample pre-treatment can reduce the matrix effect. After the three samples of tomato, cucumber, and cabbage extracted by acetonitrile, the extracts were diluted 5-fold, 10-fold, and 20-fold with PBST solution, respectively, to establish the inhibition curves and compare with the standard inhibition curves establishment by the present method. If the two curves coincide, the matrix effect is eliminated. The results showed that when the tomato and cucumber samples were diluted 5-fold and the cabbage sample was diluted 10-fold, the inhibition curves obtained overlapped with the standard inhibition curves established by the present method, indicating that the matrix effect could be eliminated.

To evaluate the accuracy of this method, this experiment measured the actual samples and added three different concentrations of fenpropathrin and procymidone standard to the samples in high, medium, and low to carry out the additive recovery validation experiments, and the experimental results obtained by the method compared with the detection results of the high-performance liquid chromatography (HPLC) method. The results are shown in Table 1. In this method, the recovery rate of fenpropathrin was 86.5~100.2%, and the recovery rate of procymidone was 85.61~102.43%. The coefficient of variation was less than 10%. The recovery rates of fenpropathrin and procymidone in HPLC were 82.13~99.63% and 88.13~102.73%, respectively. The coefficients of variation were less than 10%, indicating that the upconversion fluorescence immunoassay method established in this experiment had good accuracy and good consistency with the HPLC method. In this experimental method, the detection limits of tomato, cucumber, and cabbage were 2.5 µg/kg, 2.5 µg/kg, and 5 µg/kg, respectively, which were much lower than the national food safety standard maximum residue limits (MRLs) for pesticides [25], and this method can be used for the detection of pesticide residues of fenpropathrin and procymidone in food.

### 3.7. Actual Sample Testing

We randomly purchased five pieces of tomatoes, cucumbers, and cabbage from different online stores and numbered 15 samples from 1 to 15. We divided the usable samples of each sample into two equal parts. We used this experimental method and high-performance liquid chromatography to test the same sample, and the results are shown in Table 2. No pesticide residues of fenpropathrin and procymidone were detected in cucumbers, while one sample in tomatoes detected fenpropathrin, and procymidone was detected in two samples of cabbage. The results of the comparison between high-performance liquid chromatography and this experimental method are consistent. It is worth noting that China, EU, and Japan stipulate that the MRLs of fenpropathrin in tomato are 1 mg/kg, 0.01 mg/kg, and 2 mg/kg, respectively. Japan stipulates that the MRL of procymidone in cabbage is 0.5 mg/kg, and EU stipulates that the MRL of carbendazim in fruits and vegetables is 0.02 mg/kg. In the test samples, the putrescine of one sample residue exceeded the EU MRL, which affects the export of commodities. Tomato, cucumber, and cabbage are three kinds of staple vegetables that people often eat, and their quality and safety issues are related to people’s health. Therefore, the detection of fenpropathrin and procymidone in tomato, cucumber, and cabbage established in this experiment is of great significance.

## 4. Conclusions

This study investigated the effects of reaction temperature and time on the morphology of upconversion nanocrystals based on the high-temperature thermal decomposition method. The optimal reaction conditions were determined in order to synthesize a three-layer core-shell-shell upconversion nanomaterial with uniform size, good dispersion, and strong luminescence efficiency. Compared with its corresponding two-layer structure UCNPs, the fluorescence intensity of the material increased several times, and the core-shell-shell structure effectively suppressed fluorescence quenching. It exhibits significant advantages in terms of luminous efficiency and stability. Using this material in combination with other UCNPs with the same three-layer structure, a fluorescence immunoassay for the simultaneous detection of fenpropathrin and procymidone was established by combining the magnetic separation technique with the study object of fenpropathrin and procymidone. The results were consistent with those of the HPLC method, which proves that the method has good accuracy and practical applicability and can be used for the rapid and sensitive detection of fenpropathrin and procymidone in food.

## Figures and Tables

**Figure 1 foods-12-03445-f001:**
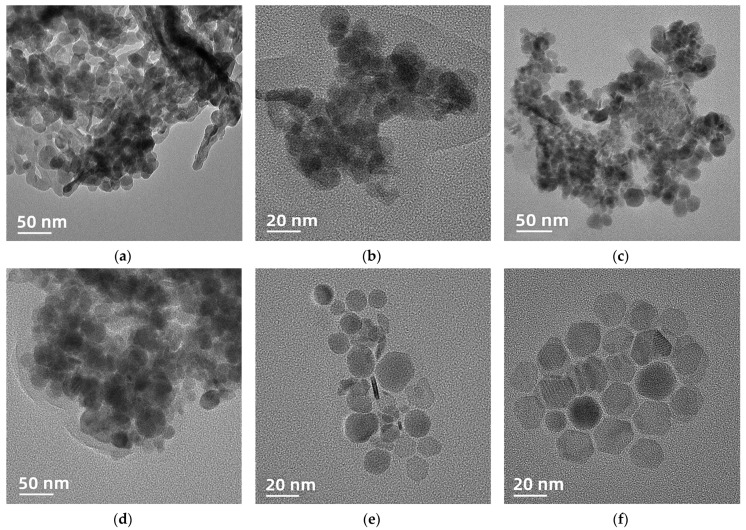
Transmission electron microscopy of NaGdF_4_ at different temperatures and times. (**a**) 1 h, 300 °C; (**b**) 1.5 h, 300 °C; (**c**) 2 h, 300 °C; (**d**) 1 h, 310 °C; (**e**) 1.5 h, 310 °C; (**f**) 2 h, 310 °C.

**Figure 2 foods-12-03445-f002:**
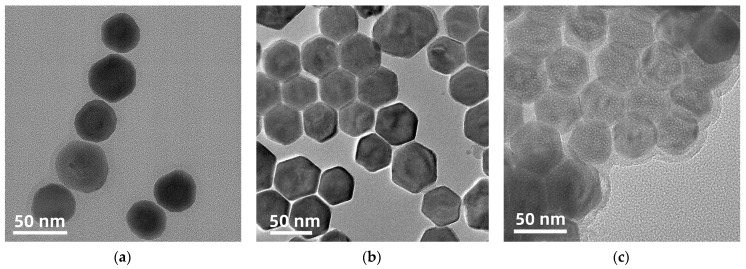
Transmission electron microscopy pictures of (**a**) NaGdF_4_ @ NaGdF_4_: Yb, Tm; (**b**) NaGdF_4_ @ NaGdF_4_: Yb, Tm @ NaGdF_4_: Yb, Nd; (**c**) NaGdF_4_ @ NaGdF_4_: Yb, Tm @ NaGdF_4_: Yb, Nd @ SiO_2_ @ NH_2_.

**Figure 3 foods-12-03445-f003:**
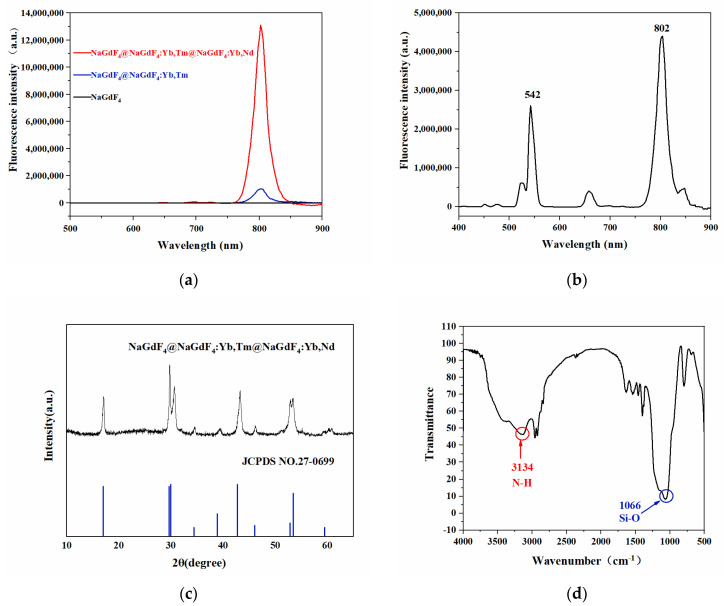
(**a**) NaGdF_4_@NaGdF_4_: Yb, Tm@NaGdF_4_: Yb, Nd fluorescence spectra. (**b**) Mixed fluorescence spectra of NaGdF_4_@NaGdF_4_: Yb, Tm@NaGdF_4_: Yb, Nd and NaYF_4_@NaYF_4_: Yb, Er@NaYF_4_. (**c**) NaGdF_4_ @ NaGdF_4_: Yb, Tm @ NaGdF_4_: Yb, Nd X-ray diffraction pattern. (**d**) Fourier transform infrared spectra of NaGdF_4_ @ NaGdF_4_: Yb, Tm @ NaGdF_4_: Yb, Nd @ SiO_2_ @ NH_2_.

**Figure 4 foods-12-03445-f004:**
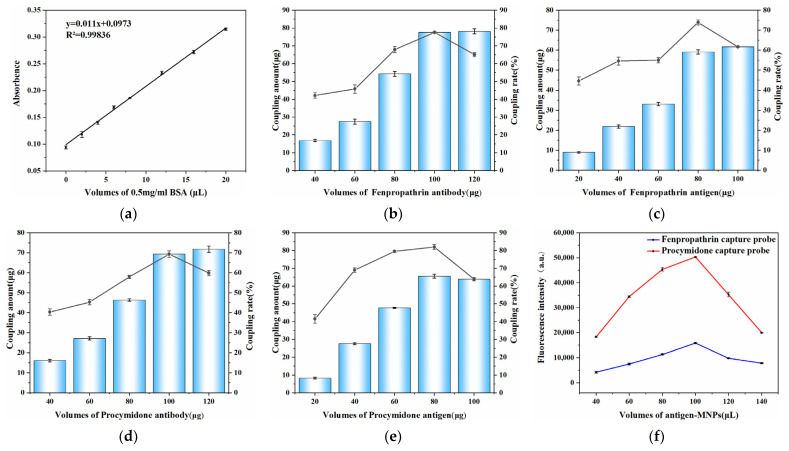
(**a**) Standard curve of BCA kit. (**b**) Optimization of fenpropathrin antibody addition. (**c**) Optimization of fenpropathrin antigen addition. (**d**) Optimization of procymidone antibody addition. (**e**) Optimization of procymidone antigen addition. (**f**) Optimization of capture probe addition.

**Figure 5 foods-12-03445-f005:**
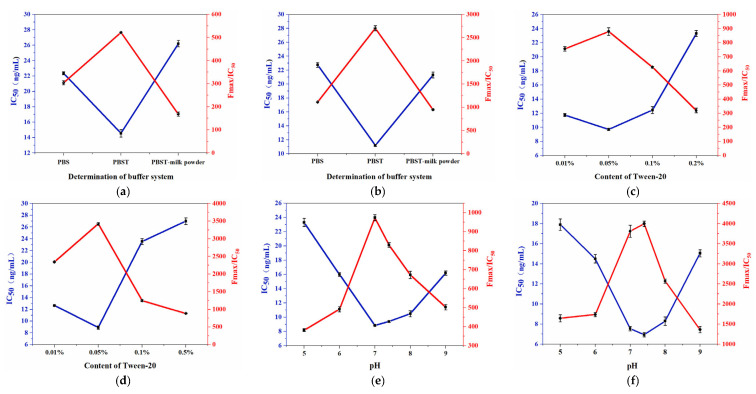
(**a**,**b**) Determination of buffer system. (**c**,**d**) Tween−20 content. (**e**,**f**) pH value. (**g**,**h**) Methanol content. (**i**) Time. (**j**) Upconversion fluorescence spectra of different concentrations of fenpropathrin and procymidone standards. (**k**) Standard curve of fenpropathrin. (**l**) Standard curve of procymidone.

**Figure 6 foods-12-03445-f006:**
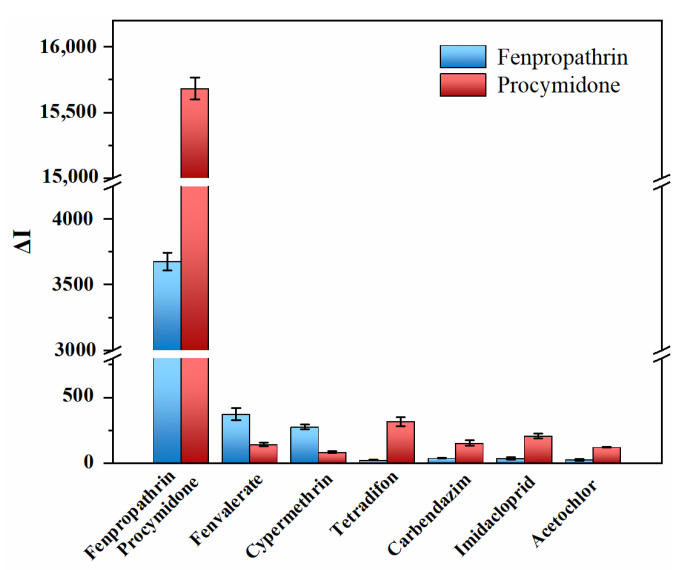
Method specificity evaluation.

**Table 1 foods-12-03445-t001:** Detection results of fenpropathrin and procymidone in vegetables by fluorescence immunoassay and high-performance liquid chromatography (*n* = 5).

Sample	Pesticide	Additive Concentration (μg/kg)	Fluorescence Immunoassay			HPLC		
			Mean ± SD (μg/kg)	Recovery Rate (%)	CV (%)	Mean ± SD (μg/kg)	Recovery Rate (%)	CV (%)
Tomato	Fenpropathrin	2.5	2.16 ± 0.12	86.50	5.5	2.22 ± 0.18	88.93	7.9
		10	9.23 ± 0.37	92.27	4.0	9.23 ± 0.32	92.25	3.4
		25	25.05 ± 0.47	100.20	1.9	24.91 ± 0.34	99.63	1.4
	Procymidone	2.5	2.30 ± 0.13	91.89	5.8	2.35 ± 0.14	93.87	6.1
		10	9.20 ± 0.24	92.02	2.6	8.99 ± 0.35	89.88	3.9
		25	24.76 ± 0.58	99.06	2.3	24.78 ± 0.30	99.11	1.2
Cucumber	Fenpropathrin	2.5	2.40 ± 0.12	96.03	5.2	2.40 ± 0.14	96.13	5.8
		10	8.97 ± 0.44	89.68	4.9	9.47 ± 0.41	94.67	4.3
		25	23.28 ± 0.68	93.11	2.9	23.12 ± 0.40	92.49	1.7
	Procymidone	2.5	2.18 ± 0.17	87.27	7.7	2.20 ± 0.16	87.87	7.2
		10	9.96 ± 0.41	99.57	4.2	9.45 ± 0.31	94.53	3.3
		25	25.61 ± 0.68	102.43	2.7	25.68 ± 0.61	102.73	2.4
Cabbage	Fenpropathrin	5	4.28 ± 0.42	89.30	8.8	4.11 ± 0.22	82.13	5.3
		20	18.24 ± 1.15	96.44	3.4	18.85 ± 0.92	94.27	4.9
		50	48.49 ± 0.87	94.03	2.1	46.49 ± 0.97	92.97	2.1
	Procymidone	5	4.28 ± 0.42	85.61	9.7	4.41 ± 0.30	88.13	6.9
		20	18.24 ± 1.15	91.24	6.3	17.71 ± 0.83	88.53	4.7
		50	48.49 ± 0.87	96.97	1.8	46.83 ± 0.98	93.67	2.1

**Table 2 foods-12-03445-t002:** Detection of fenpropathrin and procymidone in actual samples.

Sample	Fluorescence Immunoassay		HPLC	
	Fenpropathrin	Procymidone	Fenpropathrin	Procymidone
	(μg/L)	(μg/L)	(μg/L)	(μg/L)
1–4# Tomato	-	-	-	-
5# Tomato	8.7		8.2	
6–10# Cucumber	-	-	-	-
11# Cabbage	-	12.8	-	13.5
12# Cabbage	-	-	-	-
13# Cabbage	-	42.1		41.5
14–15# Cabbage	-	-	-	-

-: Indicates not detected.

## Data Availability

Data is contained within the article.

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
