# Peer review of "Core-Shell-Shell Upconversion Nanomaterials Applying for Simultaneous Immunofluorescent Detection of Fenpropathrin and Procymidone"

_foods, 2023, doi:10.3390/foods12183445_

Round 1

Reviewer 1 Report

This manuscript entitled “Core-Shell-Shell Upconversion Nanomaterials Applying for Simultaneous Immunofluorescent Detection of Fenpropathrin and Procymidone” has been written very well, the topic is valuable and interesting; however, the manuscript needs some modification:

1.      In line 95, the full word (Bicinchoninic Acid) should be given in the footnote to specify BCA kit.

2.      In section “2.4. Preparation and Optimization of Capture Probe and Signal Probe “ , lines 151 and 155,  the ultra-sonication had been used in which frequencies?

3.      In line the details of centrifugal separation must be added ( rpm, g, and time).

4.      In lines 156and 182, in which RPM the sample was shaken? Please add in text.

5.      For figure 1, if possible, similar magnifications (20 nm) should be used for different images.

6.       It would better to change the legend of figure 2 to :  Transmission electron microscopy pictures of (a)  NaGdF4  @ NaGdF4: Yb; (b)  NaGdF4 @ NaGdF4: Yb, Tm @ NaGdF4: Yb, Nd; and (c) NaGdF4 @ NaGdF4: Yb, Tm @ NaGdF4: Yb, Nd @ SiO2 @ NH2.

7.      In the text, a domain of concentrations mentioned A-B µg instead of Aµg-B µg.

Reviewer 2 Report

The study synthesized new magnetic nanoparticles modified with antibodies against two pesticides, creating capture probes for a rapid and sensitive fluorescence immunoassay. The method has the potential to be used for the rapid detection of fenpropathrin and procymidone in food.

The authors emphasized that their procedure showed good agreement with HPLC results, however, despite presenting the results in a table, they did not statistically compare the measured values and did not compare the validation parameters with other already published ones. I hope that the authors will improve this manuscript a bit.

Other doubts are:

Lines 60-62. The authors wrote the following: "Some studies have shown that the long-term accumulation of fenpropathrin and procymidone in the human body may seriously affect the nervous system and affect health [17-18]" - Please add more details. What kind of effects on human health do you mean?

Line 62-63- Authors wrote: "These two pesticide residues have become routine detections in fruits and vegetables [19]."- Please describe this routine detection method.

Line 63-66 The authors wrote the following: "As two common pesticides in pyrethroids and organochlorines, fenpropathrin and procymidone are often detected simultaneously, and the instrumental method is the main detection technique [20-21], which has high sensitivity and specificity, but expensive equipment and the need for professional operation, which makes it difficult to meet the demand for rapid on-site detection."- What technique exactly do you mean, what are the advantages, disadvantages, results of this technique?

Line 66. The authors write the following:".... but expensive equipment and the need for professional operation, which makes it difficult to meet the demand for rapid on-site detection" - What kind of equipment are the authors talking about?

Figure 7 should be deleted as Table 1 is sufficient.

Line 353-354 The authors wrote the following:" According to Fig.5(k, l), the detection limit and sensitivity of fenpropathrin were 0.114 µg/kg and 8.15 µg/kg, respectively, and the detection limit and sensitivity of procymidone were 0.082 µg/kg and 7.98 µg/kg, respectively."- Please describe the method of LOD calculation. Did you use sensitivity and detection limit as synonyms? If so, please use only one word.

Fig.5 k, l-Please add the analyte concentrations in the range of linearity.

Subchapter 3.6- Matrix effect should be described more precisely. What does it mean that "after dilution the curves coincided with the standard curves, eliminating the effect of the substrate at this time" - please show what exactly you compared?

Table 1. shows the results obtained by fluorescence immunoassay and HPLC. Please compare the results statistically.

Table 2 shows the concentrations of fenpropathrin and procymidone in real samples. The concentration of procymidone seems to be outside the linear range. Can you explain this problem?

All the figures are too small to be visible.

Author Response

请参阅附件。
